# Resistant and Relapsing Collapsing Glomerulopathy Successfully Treated with Rituximab—A Case Report

**DOI:** 10.3390/jpm12091415

**Published:** 2022-08-30

**Authors:** Nikola Zagorec, Dragan Klarić, Marta Klarić, Ivica Horvatić, Petar Šenjug, Matija Horaček, Jagoda Nikić, Danica Galešić Ljubanović, Krešimir Galešić

**Affiliations:** 1Department of Nephrology and Dialysis, Dubrava University Hospital, School of Medicine, University of Zagreb, 10000 Zagreb, Croatia; 2Department of Internal Medicine, Zadar General Hospital, 23000 Zadar, Croatia; 3School of Medicine, University of Zagreb, 10000 Zagreb, Croatia; 4Unit of Nephropathology and Electron Microscopy, Department of Pathology and Cytology, Dubrava University Hospital, School of Medicine, University of Zagreb, 10000 Zagreb, Croatia; 5Nursing School Mlinarska, University of Applied Health Sciences, 10000 Zagreb, Croatia

**Keywords:** collapsing glomerulopathy, rituximab, relapsing disease, multidrug-resistant disease, kidney biopsy

## Abstract

Collapsing glomerulopathy (CG) or collapsing focal segmental glomerulosclerosis (cFSGS) is an aggressive disease with a high tendency of progression to end-stage renal disease due to common resistance to conventional immunosuppressants. Rituximab (RTX), a monoclonal antibody against CD20 B cells, showed some benefit in the treatment of CG. We are reporting about female patients with an idiopathic form of CG presenting with nephrotic syndrome (NS) and renal insufficiency resistant to several immunosuppressive agents such as steroids (ST), calcineurin inhibitors (CNI), and cyclophosphamide (CYC). This multidrug-resistant disease responded to RTX with complete remission. Forty-four months after initial RTX administration, a relapse of CG with severe NS and acute renal insufficiency occurred. Repeated application of RTX led to complete remission again. To the best of our knowledge, we are reporting the first case of the relapsing multidrug-resistant form of CG, which responded to RTX. Current data about the treatment of CG with RTX is lacking and is based on rare case reports and small case series. Thus, our report can contribute to determining the role of RTX in the treatment of CG.

## 1. Introduction

CG is a rare and aggressive disease usually presenting with severe NS and renal insufficiency leading to end-stage renal disease [1,2,3,4]. Since the first description of CG by Weiss et al. in 1986, the etiology of CG is still not completely clarified [5]. CG may be related to human immunodeficiency virus (HIV), other viral infections, APOL1 risk variants and genetic disorders, some drugs, or may be idiopathic [1]. Due to common resistance to conventional immunosuppressive (IS) agents such as ST and CNI, treatment of CG remains challenging [2,4,6]. Currently, there are no recommendations for the treatment of steroid-resistant and CNI-resistant CG. The number of articles reporting favorable effects of off-label RTX use in minimal change disease (MCD), primary FSGS, and CG grew over the last decade, although randomized controlled trials are still missing [7,8,9,10,11]. RTX is a monoclonal anti-CD20 antibody with a direct B cell-depleting effect and other immune and nonimmune effects, which result in antiproteinuric effects in renal diseases [12]. Successful use of RTX in the treatment of CG was first reported in the pediatric population [7,8] and then in adults [9,10,11]. According to available literature, there are no reports of treatment of the relapsing form of CG, especially not with RTX.

## 2. Case Description

A female born in 1950 presented in February 2010 with progressive lower limb swelling for the duration of several months. Past medical history included normal birth weight, transitory proteinuria of unknown amount and etiology during pregnancy in 1974 (accompanied by lower limb and periorbital edema), arterial hypertension, and mixed hyperlipidemia for 7 years. She provided no history of recent infections, arthralgia, skin lesions, new medication intake, or vaccination. Her daily medications included telmisartan (80 mg/day) and simvastatin (80 mg/day). Family history was negative for kidney diseases. On examination, she had bilateral leg edema, she was normotensive (120/80 mmHg), and other vital signs were within normal limits, as well as an examination of the head and neck, cardiovascular system, chest, abdomen, and arms. Diagnostic workup revealed NS with 24 h proteinuria of 5.8 g/day (Table 1 shows relevant laboratory data). Complete immunology, screening for malignant diseases, and serology for HIV, hepatitis B and C virus, parvovirus B19, cytomegalovirus, and Epstein-Barr virus were negative.

Given the NS and unclear diagnosis, the first kidney biopsy was performed. On light microscopy, 3 out of 19 glomeruli were globally sclerosed, and 2 showed segmental glomerular collapse and podocyte hyperplasia. Interstitial fibrosis and tubular atrophy (IFTA) were very mild (5%), and mild to moderate arteriolar hyalinosis with mild fibrointimal thickening of larger arteries was present. Immunofluorescence (IF) was negative. On electron microscopy (EM), podocytes showed diffuse foot process effacement, and the glomerular basement membrane thickness was within referral values for our renal pathology laboratory. The histological finding (Figure 1A and 1B) in correlation with clinical features were consistent with CG.

After an initial four-month course of prednisone (1 mg/kg of body weight) without clinical response, cyclosporine (CsA) was added to the treatment in a dose of 3 mg/kg, divided into two daily doses (targeted blood level of CsA was 100–400 μg/L 12 h after last oral dose). Medication adherence was checked by regular CsA level measurement (every follow-up visit). A 12-month course of CsA and ST was discontinued due to ineffectiveness and worsening of proteinuria (up to 10 g/day). A second kidney biopsy was performed in November 2012 because of ST and CNI resistance. On light microscopy, 5 out of 25 glomeruli were globally sclerosed, 4 glomeruli showed segmental sclerosis with extensive capillary tuft collapse and podocyte hyperplasia in one glomerulus (Figure 1C). IFTA progressed to 30%, and vascular changes were the same as in the first biopsy. IF showed weak granular mesangial focal segmental positivity for IgM, and EM was without immune deposits. There was diffuse podocyte foot processes effacement. Extensive clinical reevaluation was performed without any new clinical condition revealed. At this moment, kidney function was still preserved (Table 1). Treatment with dual RAAS blockade (perindopril and telmisartan), methylprednisolone (MP), and cyclophosphamide (CYC, 2 mg/kg in two daily doses) was started. CYC was discontinued after 3 months due to ineffectiveness and leukopenia (2.3 × 10^9^/L). In August 2013, tacrolimus (TAC) was started (0.08 mg/kg in two daily doses), and the low dose of MP was continued. The targeted blood level of TAC was 5–20 μg/L 12 h after the last oral dose, and medication adherence was checked by regular TAC level measurement (every follow-up visit). After 13 months of treatment, no benefit of TAC therapy was observed, and it was discontinued. During the TAC course, she experienced an episode of herpes zoster infection, which was treated with peroral acyclovir.

In November 2014, she was admitted to the hospital due to anasarca, pericardial effusion, acute renal insufficiency (serum creatinine, sCr 308 μmol/L), and massive proteinuria of 16.7 g/day. A third kidney biopsy was performed. On light microscopy, 13 out of 27 glomeruli were globally sclerosed, and 3 glomeruli showed capillary tuft collapse and podocyte hyperplasia. IFTA involved 40% of the sample. The vascular changes, IF, and EM findings were similar to previous biopsies. After initial symptomatic treatment with albumin infusion and diuretics, treatment with RTX was started in December 2014. RTX was applied in two fixed doses of 500 mg in a two-week period. MP in a dose of 0.5 mg/kg was continued, and acyclovire, diflucane, and sulfomethoxazole-trimetroprim were prescribed prophylactically. Six months after RTX administration, proteinuria and sCr were 1.48 g/day and 112 μmol/L, respectively, and additional two doses of 500 mg of RTX in a two-week period were applied. B cell (CD19) count was not measured at this time. The time to partial and complete remission after initial RTX administration was 3 and 11 months, respectively. After complete resolution of proteinuria, kidney function remained moderately reduced (eGFR 45 mL/min/1.73 m^2^), and the MP dose was tapered to 4 mg per day and continued because of arthralgia.

Forty-four months after initial RTX administration, in August 2018, a relapse of NS occurred (data shown in Table 1). RTX infusions were repeated (two doses of 500 mg in a two-week period followed by a single dose of 500 mg six months later) in treatment of disease relapse, which again resulted in complete remission of the disease. MP dose was also escalated to 32 mg/day and tapered over four months to a daily dose of 4 mg, which was further continued. Figure 2 shows serum albumin, sCr, and 24 h proteinuria trends after RTX administration. CD19 B cell count was 0 in January 2020 and 33 × 10^6^/L in October 2021 (normal value 100–500 × 10^6^/L). On the last visit (December 2021), complete remission of proteinuria (0.16 g/day) still persisted, and eGFR was 43 mL/min/1.73 m^2^. During follow-up, there were no adverse effects potentially related to RTX.

## 3. Discussion

Treatment with RTX led to complete and long-lasting remission in our patient with a resistant form of CG. After relapse occurred, remission was re-achieved with repeated RTX application. Some previous reports showed the success of the RTX application in patients with CG in remission induction [7,8,9,10,11]. We report the first case of successful treatment of relapsing CG with RTX after initial resistance to multiple immunosuppressants and responsiveness to RTX. According to KDIGO 2021, our patient had CNI-resistant steroid-resistant NS or, moreover, multidrug-resistant NS for failing to respond to steroids, CNI (both CsA and TAC), and CYC [13].

The use of RTX is well established in numerous immune-mediated diseases such as membranous nephropathy, ANCA-associated vasculitis, systemic lupus, etc., but its use in primary FSGS and CG is still off-label. Some observational studies have confirmed the efficacy of RTX in MCD, primary FSGS, and CG [13]. The first report of the successful use of RTX in the treatment of adult-onset CG was provided by Ramachandran et al. in a patient with CG resistant to steroids and conventional IS agents (CNI, mycophenolate mofetil, and CYC) [9]. The largest case series of 8 adult patients with CG who received RTX was reported by the same group in 2021. Five patients responded to RTX, and none of them experienced relapse after remission achieved with RTX [11]. One of the reported cases with a favorable effect of RTX was actually the same patient reported by Ramachandran et al. in 2013 [9,11]. Of patients who responded to RTX, two were steroid and CNI-dependent, one was steroid-resistant and CNI-dependent, and two were steroid and CNI-resistant. It is not clear what predicts a suitable response to RTX in patients with primary FSGS and CG, although patients who responded to steroids and CNI seem to have a better remission rate with RTX compared to those with a resistant form of the disease [12].

The effect of RTX is mostly based on depletion of B cells via binding to CD20, but also other mechanisms such as modifying effect on T cells, depletion of a small number of CD20-positive T cells, and direct regulation of podocyte function are important [12]. In our case, the RTX dosage regimen was tailored by clinical response, and, after two initial infusions of 500 mg of RTX in two-week periods, an additional infusion was given after 6 months because of incomplete resolution of proteinuria. This resulted in complete remission of the disease, both in the initial and treatment of relapse. We chose a reduced dosage of RTX in our case because of the high risk of infective complications of therapy due to previous infective complications (herpes zoster), CYC-induced leukopenia, which demanded discontinuation of IS therapy, and a high cumulative dose of previously applied IS agents. The cumulative dose of RTX applied in our case was 3500 mg (2000 mg in initial treatment and 1500 mg in relapse treatment). Unfortunately, in our case, CD19 B cell count was not measured at the beginning of treatment with RTX, but later analysis (10 months after the last RTX infusion) showed total depletion of CD19-positive cells with partial recovery 30 months after the last infusion. The role of CD19^+^ cells measuring in predicting response to RTX and relapse occurrence in FSGS and CG is still unclear [12]. Colluci et al. showed that only switched memory B cells (CD19^+^CD27^+^IgM^−^IgD^−^) play a role in the recurrence of NS in patients with idiopathic NS treated with RTX. Reconstitution of switched memory B cells was associated with time to relapse after RTX treatment, and their recovery at 9 months after initial RTX treatment was strongly predictive of relapse of idiopathic NS [14]. The optimal dosing protocol of RTX in primary FSGS and CG is still unknown, as well as the application of subsequent doses of RTX for consolidation and long-term remission [12]. The retrospective study of Lin et al. showed the benefit of consolidation therapy with RTX in patients with FSGS who achieved remission of disease, but in this cohort, there were no patients with CG [15]. Besides the excellent effect on remission induction, an additional benefit of RTX use in our case is an important steroid-sparing effect. After RTX application, the steroid (MP) dose was tapered to a minimal dose of 4 mg kept because of arthralgia.

It is important to emphasize the role of repeated kidney biopsy in conditions of no responsiveness to conventional IS agents, such as in our case. Although basic diagnosis remained unchanged, repeated biopsy is very important in the assessment of chronic histological changes (IFTA and global glomerulosclerosis), which could explain incomplete eGFR recovery after RTX use despite complete resolution of proteinuria.

## 4. Conclusions

RTX seems to be a reasonable treatment option in the idiopathic form of adult-onset CG resistant to steroids and conventional IS agents, and we showed the favorable effect of repeated application of RTX in the treatment of relapsing disease. Further clinical trials are warranted to define the role of RTX in the treatment of CG as well as dosage regimen and predictors of the patient’s responsiveness to RTX.

## Figures and Tables

**Figure 1 jpm-12-01415-f001:**
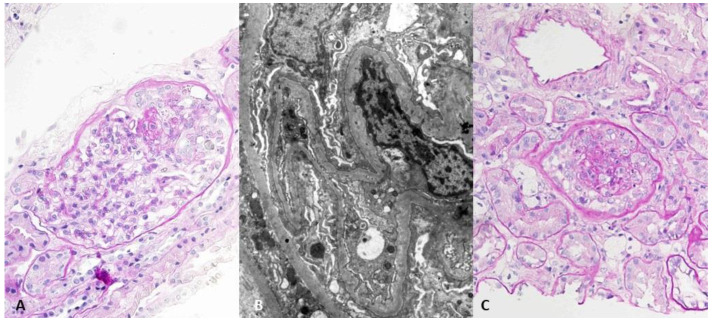
(**A**) Light microscopy of the first biopsy—glomerulus with segmental tuft collapse and podocyte hyperplasia. PAS, ×400. (**B**) Transmission electron microscopy of the first biopsy—diffuse podocyte foot processes effacement. ×8000. (**C**) Light microscopy of the second biopsy—glomerulus with tuft collapse and podocyte hyperplasia. PAS, ×400.

**Figure 2 jpm-12-01415-f002:**
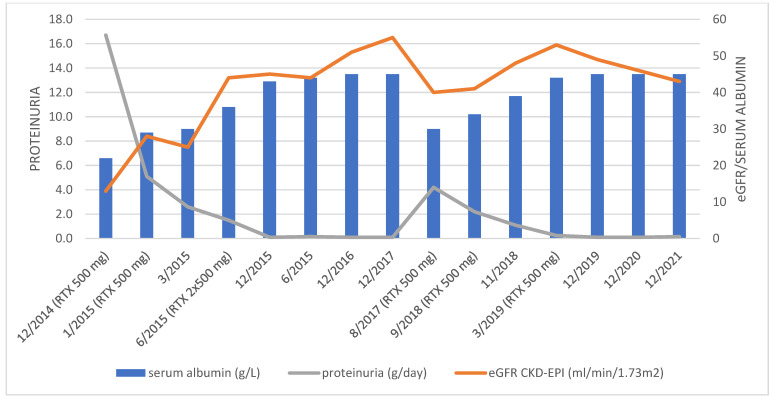
Trends of serum albumin, eGFR, and 24 h proteinuria during the RTX treatment.

**Table 1 jpm-12-01415-t001:** Important laboratory parameters during treatment with RTX in case of CG.

Laboratory Parameter	February 2010 (Time of Diagnosis)	November 2012 (CYC ^1^ Initiation)	November 2014 (First RTX ^2^ Application)	August 2018 (First Relapse after RTX)	December 2021 (Last Visit)
Hemoglobin (g/L)	124	117	127	126	132
Leukocytes (×10^6^/L)	9.2	5.8	7.7	5.7	9.2
Serum creatinine (μmol/L)	56	78	308	99	111
eGFR CKD-EPI ^3^ (ml/min/1.73 m^2^)	97	70	13	50	43
Serum albumin (g/L)	23	24	19	30	42
Urea (mmol/L)	5.2	6.4	35.2	10.4	n/a
Total cholesterol (mmol/L)	10	9.8	10.9	9.7	5.75
LDL (mmol/L)	5.77	6.4	n/a	6.89	3.26
Triglycerides (mmol/L)	1.62	2.4	4.47	1.79	0.99
IgG (g/L)	n/a	2.58	n/a	n/a	n/a
C3 (g/L, ref. 0.9–1.8)	n/a	1.35	n/a	n/a	n/a
C4 (g/L, ref. 0.1–0.4)	n/a	0.39	n/a	n/a	n/a
Complete immunology	Negative	Negative	Negative	Negative	n/a
Viral hepatitis and HIV ^4^ serology	Negative	Negative	Negative	Negative	n/a
24 h proteinuria (g/day)	5.08	7.54	16.7	4.2	0.16
Urine sediment, erythrocytes (per HPF ^5^)	8	8–10	14	6	0

^1^ cyclophosphamide; ^2^ rituximab; ^3^ Chronic Kidney Disease Epidemiology Collaboration; ^4^ human immunodeficiency virus; ^5^ high-powered field.

## Data Availability

Not applicable.

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
