# Peer review of "Resistant and Relapsing Collapsing Glomerulopathy Successfully Treated with Rituximab—A Case Report"

_jpm, 2022, doi:10.3390/jpm12091415_

Round 1

Reviewer 1 Report

Cumulative dose of rituximab should be mentioned. A delayed reconstitution of switched memory B cells may be useful to predict relapses in these patients and should be briefly discussed.

Author Response

Point 1: cumulative dose of rituximab should be mentioned.

Response 1: Thank you very much for your comment. Cumulative dose of rituximab was 3500 mg (2000 mg for initial treatment and 1500 mg for relapse treatment).

We added this in "Discussion" (please see page 5, line 163) explaining also reduced dosage of RTX.

Point 2: A delayed reconstitution of switched memory B cells may be useful to predict relapses in these patients and should be briefly discussed.

Response 2: Thank you very much for your excellent comment and suggestion. 

To discuss about role of switched memory B cells in relepse of nephrotic syndrome after treatment with rituximab, we added a new reference (No 14: Colucci, M.; Carsetti, R.; Cascioli, S.; Casiraghi, F.; Perna, A.; Rava, L.; Ruggiero, B., Emma, F.; Vivarelli, M. B Cell Reconstitution after Rituximab Treatment in Idiopathic Nephrotic Syndrome. J Am Soc Nephrol. 2016, 27, 1811-1822.).

As you can see in "Discussion", we added a brief discussion about role of memory B cells in following sentences: "Colluci et al showed that only switched memory B cells (CD19+CD27+IgM-IgD-) play role in recurrence of NS in patients with idiopathic NS treated with RTX. Reconstitution of switched memory B cells was asocciated with time to relapse after RTX treatment and their recovery at 9 months after RTX treatment was strongy predictive of relapse of idiopathic NS [14]. " (Please, see page 5, line 169)

Best regards,

Authors

Reviewer 2 Report

Very interesting case 

I have few points 

For cyclosporine and tacrolimus was there any issues  regarding compliance and did you check any  

Did you check through level and what was you targeted 

level 

For rituxmab you give two fixed doses of 500 mg but 

i believe that rituxmab dose should be 375 mg per m2

Do you think there is any role for plasma exchange 

in collapsing FSGS

Author Response

Point 1: For cyclosporine and tacrolimus was there any issues regarding compliance and did you check any? Did you check through level and what was you targeted level?

Response 1: To the our best knowledge, compliance to drug (cyclosporin and tacrolimus) taking in case of our patient was good. We were checking drug level all the time while treatment with cyclosporin and tacrolimus (at every follow up drug level was checked). Almost at every measurement, drug level was in targeted refferal interval. For cyclosporin targeted level was 100 - 400 ug/L (12 h after last oral dose) and for tacrolimus target level was 5 - 20 ug/L (12 h after last oral dose). 

We added response to your comment in manuscript (please see "Case Description", page 3, line 81 and 96).

Point 2: For rituxmab you give two fixed doses of 500 mg but i believe that rituxmab dose should be 375 mg per m2.

Response 2: Thank you for your comment/observation. You are absolutely right and rituximab should be given in two doses of 1000 mg (2 weeks apart) - rheumatic protocol or as 4- weekly doses of 375 mg/m2 - lymphoma protocol. In our case, we decided to apply reduced dose of RTX (500 mg) according so called "rheumatic protocol" because of high risk of infective complications of RTX therapy due to: previous infective complication (herpes zooster during tacrolimus treatment), cyclophosphamide-induced leukopenia which demanded therapy discontinuiation and high cumulative dose of previous applied immunosuppressive agents (steroids, CYC, CsA, TAC). 

We added our explanation of RTX dosage regimen in "Discussion" (please see page 5, line 159).

Point 3: Do you think there is any role for plasma exchange in collapsing FSGS?

Response 3: Thank you for your excellent question. At our Nephrology Department there were more than 2500 native kidney biopsies performed and more than 100 patients with primary form of FSGS was diagnosed and treated. We used plasma exchange only in one case of female patient with severe form of cellular FSGS. She presented with severe nephrotic syndrome after delivery and acute renal insufficiency with rapid tendency of renal function worsening. After unsuccessful treatment with steroids, CYC and RTX, we started plasma exchange, but without any access. She developed end-stage renal disease within several months after clinical presentation. Unfortunately, we do not have any experience with plasma exchange in treatment of collapsing glomerulopathy. By our opinion, plasma exchange is therapeutic option for relapse of collapsing glomerulopathy after kidney transplantation and, maybe, one of salvage therapeutic options in resistant forms of collapsing glomerulopathy with high tendency of progression to end-stage renal disease. However, due to different pathogenesis of CG and other forms of primary FSGS, we can not be sure what to expect from plasma exchange in relapsing CG after kidney transplantation. 

We hope we answered your question.  

Best regards, 

Authors